# ESTIMATING TREATMENT EFFECTS USING NEUROSYMBOLIC PROGRAM SYNTHESIS

## ABSTRACT

Estimating treatment effects from observational data is a central problem in causal inference. Methods to solve this problem exploit inductive biases and heuristics from causal inference to design multi-head neural network architectures and regularizers. In this work, we propose to use neurosymbolic program synthesis, a data-efficient, and interpretable technique, to solve the treatment effect estimation problem. We theoretically show that neurosymbolic programming can solve the treatment effect estimation problem. By designing a Domain Specific Language (DSL) for treatment effect estimation problem based on the inductive biases used in literature, we argue that neurosymbolic programming is a better alternative to treatment effect estimation than traditional methods. Our empirical study reveals that our method, which implicitly encodes inductive biases in a DSL, achieves better performance on benchmark datasets than the state-of-the-art methods.

## 1 INTRODUCTION

Treatment effect (also referred to as causal effect) estimation estimates the effect of a treatment variable on an outcome variable (e.g., the effect of a drug on recovery). Randomized Controlled Trials (RCTs) are widely considered as the gold standard approach for treatment effect estimation (Chalmers et al., 1981; Pearl, 2009). In RCTs, individuals are randomly split into the *treated* group and the *control (untreated)* group. This random split removes the spurious correlation between treatment and outcome variables before the experiment so that estimated treatment effect is unbiased. However, RCTs are often: (i) unethical (e.g., in a study to find the effect of smoking on lung disease, a randomly chosen person cannot be forced to smoke), and/or (ii) impossible/infeasible (e.g., in finding the effect of blood pressure on the risk of an adverse cardiac event, it is impossible to intervene on the same patient with and without high blood pressure with all other parameters the same) (Sanson-Fisher et al., 2007; Carey & Stiles, 2016; Pearl et al., 2016). These limitations leave us with observational data to compute treatment effects.

Observational data, similar to RCTs, suffers from *the fundamental problem of causal inference* (Pearl, 2009), which states that for any individual, we cannot observe all potential outcomes at the same time (e.g., once we record a person's medical condition after taking a medicinal drug, we cannot observe the same person's medical condition with an alternate placebo). Observational data also suffers from selection bias (e.g., certain age groups are more likely to take certain kinds of medication compared to other age groups) (Collier & Mahoney, 1996). For these reasons, estimating unbiased treatment effects from observational data can be challenging (Hernan & Robins, 2019; Farajtabar et al., 2020). However, due to the many use cases in the real-world, estimating treatment effects from observational data is one of the long-standing central problems in causal inference (Rosenbaum & Rubin, 1983; 1985; Brady et al., 2008; Morgan & Winship, 2014; Shalit et al., 2017; Yoon et al., 2018; Shi et al., 2019; Yao et al., 2018; Zhang et al., 2021).

Earlier methods that estimate treatment effects from observational data are based on matching techniques that compare data points from treatment and control groups that are similar w.r.t. a metric (e.g., Euclidean distance in nearest-neighbor matching, or propensity score in propensity score matching) (Brady et al., 2008; Morgan & Winship, 2014). Recent methods exploit inductive biases and heuristics from causal inference to design multi-head neural network (NN) models and regularizers (Hill, 2011; Farajtabar et al., 2020; Shi et al., 2019; Schwab et al., 2020; Chu et al., 2020; Shalit et al., 2017; Alaa & van der Schaar, 2017; Yoon et al., 2018; Bica et al., 2020; Künzel et al., 2019; Chernozhukov et al., 2018; Yao et al., 2018; Zhang et al., 2021). Multi-head NN models are typically used when treatment variables are single-dimensional

and categorical (Shi et al., 2019; Shalit et al., 2017; Farajtabar et al., 2020; Schwab et al., 2020), and regularizers therein enforce constraints such as controlling for propensity score instead of pre-treatment covariates, i.e. covariates that are not caused by the treatment variable in the underlying causal data generating graph (Shi et al., 2019; Rosenbaum & Rubin, 1983).

However, each such model is well-suited to a certain kind of causal graph, and may not apply to all causal data generating processes. For example, as shown in Fig 1, CFRNet (Shalit et al., 2017), a popular NN-based treatment estimation model, controls pre-treatment covariates using a regularizer based on an Integral Probability Metric (IPM). It requires the representations of non-treatment covariates (denoted as $x$ in Fig 1) with and without treatment to be similar. This is relevant for causal model A in the figure, but does not work for causal model B, where non-treatment covariates are caused by the treatment $t$ and hence could vary for different values of $t$. One would ideally need a different regularizer or architecture to address causal model B (the same observation holds for TARNet (Shalit et al., 2017) too). In practice, one may not be aware of the underlying causal

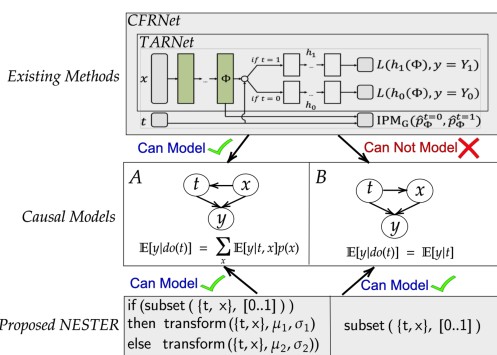

Figure 1: IPM regularization in CFRNet (note that CFRNet is a combination of a simple two-head TARNet with IPM regularization) controls for covariates $x$ which may lead to incorrect treatment effect w.r.t. causal model B. However, NESTER learns to synthesize different estimators for the two causal models.

model, making this more challenging. In this work, we instead propose to use a neurosymbolic program synthesis technique to compute treatment effect, which does not require such explicit regularizers or architecture redesign for each causal model. Such a technique learns to *automatically synthesize differentiable programs* that satisfy a given set of input-output examples (Shah et al., 2020; Parisotto et al., 2016), and can hence learn the sequence of operations to estimate treatment effect for this set. We call our method as the NEuroSymbolic Treatment Effect EstimatoR (or NESTER). Neurosymbolic program synthesis is known to have the flexibility to synthesize different programs for different data distributions to optimize a performance criterion, while still abiding by the inductive biases studied in treatment effect estimation literature (see Sec 4.1 for more details). To describe further, one could view CFRNet/TARNet as implementing one `if − then − else` program primitive with its two-headed NN architecture. NESTER will instead automatically synthesize the sequence of program primitives (from a domain-specific language of primitives) for a given set of observations from a causal model, and can thus generalize to different distributions.

Program synthesis methods, in general, enumerate a set of programs and select (from the enumeration) a set of feasible programs that satisfy given input-output examples so that the synthesized programs generalize well to unseen inputs (see Appendix for an example) (Biermann, 1978; Gulwani, 2011; Parisotto et al., 2016; Valkov et al., 2018; Shah et al., 2020). Usually, a Domain-Specific Language (DSL) (e.g., a specific *context-free grammar*) is used to synthesize relevant programs for a given domain and task. Recently, various NN-based techniques have been proposed to perform neurosymbolic program synthesis (Parisotto et al., 2016; Valkov et al., 2018; Gaunt et al., 2017; Bošnjak et al., 2017). We use the neurosymbolic program synthesis paradigm where each program primitive (e.g., `if − then − else`, $\alpha_1 + \alpha_2$) is a differentiable module (Parisotto et al., 2016; Shah et al., 2020). Such *differentiable programs* simultaneously optimize program primitive parameters while learning the overall program structure and flow. The set of possible programs that can be synthesized using a DSL is often large (Parisotto et al., 2016). Many methods have been proposed to search through the vast search space of programs efficiently (Gulwani et al., 2012; Parisotto et al., 2016; Valkov et al., 2018; Shah et al., 2020). We use *Neural Admissible Relaxation* (Shah et al., 2020) in this work, which uses neural networks as relaxations of partial programs while searching the program space using informed search algorithms such as $A^*$ (Hart et al., 1968). The final program can be obtained by training using gradient descent algorithms. Our key contributions are:

- We study the use of neurosymbolic program synthesis as a practical approach for solving treatment effect estimation problems. To the best of our knowledge, this is the first such effort that applies neurosymbolic program synthesis to estimate treatment effects.

- We propose a Domain-Specific Language (DSL) for treatment effect estimation, where each program primitive is motivated from basic building blocks of models for treatment effect estimation in literature.

- We theoretically show that our neurosymbolic program synthesis approach can approximate a continuous function upto an arbitrary precision. This result enables us to solve the treatment effect estimation problem by assuming a continuous function relating the treatment and outcome variables.
- We perform comprehensive empirical studies on multiple benchmark datasets (including additional results in the Appendix) where we outperform existing state-of-the-art models. We also show the interpretability of such a neurosymbolic approach on synthetic as well as real-world datasets, thus highlighting the usefulness of our approach over traditional treatment effect estimation methods.

## 2 RELATED WORK

**Traditional Methods for Treatment Effect Estimation.** Early methods of treatment effect estimation from observational data are largely based on matching techniques (Brady et al., 2008; Morgan & Winship, 2014; Stuart, 2010) where similar data points in treatment and control groups are compared using methods such as nearest neighbor matching and propensity score matching to estimate treatment effects. In nearest neighbor matching (Stuart, 2010), for each data point in the treatment group, the nearest points from the control group w.r.t. Euclidean distance are identified, and the difference in potential outcomes between treatment and corresponding control data points is estimated as the treatment effect. In propensity score matching (Rosenbaum & Rubin, 1983), a model is trained to predict the treatment effect value using all data points from both treatment and control groups. Using this model, points from treatment and control groups that are close w.r.t. the model's output are compared, and the difference in potential outcomes of these points is estimated as treatment effect. However, such matching techniques are known to not scale to high-dimensional or large-scale data (Abadie & Imbens, 2006).

Another family of methods estimates treatment effects using the idea of backdoor adjustment (Pearl, 2009; Rubin, 2005). Under the assumption of availability of a sufficient adjustment set (Pearl, 2009), these models rely on fitting conditional probabilities given the treatment variable and a sufficient adjustment set of covariates. However, such models are known to suffer from high variance in the estimated treatment effects (Shalit et al., 2017). Covariate balancing is another technique to control for the confounding bias to estimate treatment effects. Weighting techniques perform covariate balancing by assigning weights to each instance based on various techniques (e.g., weighting each instance using propensity score in the inverse probability weighting technique) (Rosenbaum & Rubin, 1983; Assaad et al., 2021; CRUMP et al., 2009; L & T, 2013; Diamond & Sekhon, 2013; Li & Fu, 2017). As noted in (Assaad et al., 2021), such methods face challenges with large weights and high-dimensional inputs. Besides, leveraging the success of learning-based methods has delivered significantly better performance in recent years.

**Learning-based Methods for Treatment Effect Estimation.** Recent methods to estimate treatment effects have largely been based on multi-headed NN models equipped with regularizers (Hill, 2011; Farajtabar et al., 2020; Shi et al., 2019; Schwab et al., 2020; Chu et al., 2020; Shalit et al., 2017; Yoon et al., 2018; Bica et al., 2020). To find treatment effects under multiple treatment values and continuous dosage for each treatment, (Schwab et al., 2020) devised an NN architecture with multiple heads for multiple treatments, and multiple sub-heads from each of the treatment-specific heads to model (discretized) dosage values. CFRNet (Shalit et al., 2017) proposed a two-headed NN architecture with a regularizer that forced representations of treatment and control groups to be close to each other, in order to adjust for confounding features before forwarding the representation to treatment-specific heads. Extending CFRNet architecture, (Farajtabar et al., 2020) proposed an additional regularizer to adjust for confounding by forcing both treatment-specific heads to have same baseline outcomes (i.e., for any data point, both treatment-specific heads should output same value). In Dragonnet (Shi et al., 2019), along with two heads for predicting treatment-specific (potential) outcomes, an additional head to predict treatment value was also used; this allowed pre-treatment covariates to be used in predicting potential outcomes. Assuming that potential outcomes are strongly related, (Curth & van der Schaar, 2021) proposed techniques that improve existing models using the structural similarities between potential outcomes. All of these methods, however, have a fixed architecture design and can hence address observational data from certain causal models. Our approach is also NN-based but uses a neurosymbolic approach to automatically synthesize an architecture (or a flow of program primitives), thereby providing it a capability to work across observational data from different causal models conveniently.

Generative Adversarial Networks (GANs) (Goodfellow et al., 2014) have also been used to learn the interventional distribution (Yoon et al., 2018; Bica et al., 2020) from observed data in both

categorical and continuous treatment variable settings to estimate treatment effects. By disentangling confounding variables from instrumental variables, (Zhang et al., 2021) proposed a variational inference method for treatment effect estimation that uses only confounding variables. However, generative modeling requires a large amount of data to be useful, which is often not practical in treatment effect estimation tasks. (Yao et al., 2018) proposed a method to learn representations by leveraging local similarities and thereby estimate treatment effect. Ensemble models such as causal forests (Wager & Athey, 2018), and Bayesian additive regression trees (Chipman et al., 2010) have also been considered for interval estimation. As stated earlier, our work is however very different from these efforts, and seeks to build a flexible yet powerful framework for treatment effect estimation using neurosymbolic program synthesis.

**Neurosymbolic Program Synthesis.** Program synthesis, *viz.* automatically learning a program that satisfies a given set of input-output examples (Biermann, 1978; Gulwani, 2011; Parisotto et al., 2016; Valkov et al., 2018; Shah et al., 2020), has been shown to be helpful in diverse tasks such as low-level bit manipulation code (Solar-Lezama et al., 2005), data structure manipulations (Solar Lezama, 2008), and regular expression-based string generation (Gulwani, 2011). For each task, a specific DSL is used to synthesize programs. Even with a small DSL, the number of programs that can be synthesized is very large. Several techniques such as greedy enumeration, Monte Carlo sampling, Monte Carlo tree search, evolutionary algorithms, and recently, node pruning with neural admissible relaxation have been proposed to efficiently search for optimal programs from a vast search space (Gulwani et al., 2012; Parisotto et al., 2016; Valkov et al., 2018; Shah et al., 2020). We use the idea of node pruning with neural admissible relaxation (Shah et al., 2020) in this work as it gives near-optimal solutions with fast convergence. This is the first use of neurosymbolic program synthesis for treatment effect estimation, to the best of our knowledge.

## 3 BACKGROUND AND PROBLEM FORMULATION

**Treatment Effect Estimation:** Let $\mathcal{D} = \{(\mathbf{x}_i, t_i, y_i)\}_{i=1}^n$ be an observational dataset of $n$ triplets. Each triplet $(\mathbf{x}_i, t_i, y_i)$ is a sample drawn from the true data distribution $p(\mathbf{X}, T, Y)$, where $\mathbf{X}$, $Y$ and $T$ are the corresponding random variables (described herein). $\mathbf{x}_i \in \mathbb{R}^d$ denotes the $d-$ dimensional covariate vector, $t_i \in \mathbb{R}$ denotes the treatment value ($t_i$ is not a part of $\mathbf{x}_i$), and $y_i \in \mathbb{R}$ denotes the corresponding outcome. To explain treatment effects, consider a simple setting where treatment is binary-valued i.e., $t \in \{0, 1\}$. For the $i^{th}$ observation, let $Y_i^0$ denote the true potential outcome under treatment $t_i = 0$ and $Y_i^1$ denote the true potential outcome under treatment $t_i = 1$. Because of *the fundamental problem of causal inference*, we observe only one of $Y_i^0, Y_i^1$ for a given $[t_i; \mathbf{x}_i]$. The observed outcome $y_i$ can be expressed in terms of true potential outcomes as: $y_i = t_i Y_i^1 + (1 - t_i) Y_i^0$. One of the goals in treatment effect estimation from observational data is to learn the estimator $f(\mathbf{x}, t)$ such that the difference between estimated potential outcomes (i.e., under $t = 1$ and $t = 0$): $f(\mathbf{x}_i, 1) - f(\mathbf{x}_i, 0)$ is as close as possible to the difference in true potential outcomes: $Y_i^1 - Y_i^0 \; \forall i$. This difference for a specific individual $i$ is known as *Individual Treatment Effect (ITE)* (Pearl, 2009). Extending the discussion on *ITE* to an entire population, our goal is to estimate the *Average Treatment Effect (ATE)* of the treatment variable $T$ on the outcome variable $Y$ which is defined as:

$$ATE_T^Y = \mathbb{E}[Y|do(T = 1)] - \mathbb{E}[Y|do(T = 0)] \tag{1}$$

where the $do(.)$ notation denotes external intervention to the treatment variable (Pearl, 2009), i.e. $\mathbb{E}[Y|do(T = t)]$ refers to the expected value of the outcome $Y$ when every individual in the population is administered with the treatment $t$. (Note that if treatment is not binary-valued, treatment effects are calculated w.r.t. a baseline treatment value (Pearl, 2009), and the right term in Eqn 1 would compute the interventional expectation at the baseline.) Assuming $\mathbf{X}$ satisfies the backdoor criterion relative to the treatment effect of $T$ on $Y$ (Pearl, 2009), we can write $\mathbb{E}[Y|do(T = t)] = \mathbb{E}_{\mathbf{x} \sim \mathbf{X}} [\mathbb{E}[Y|T = t, \mathbf{X} = \mathbf{x}]]$. Using this, a simple technique to estimate $\mathbb{E}[Y|T = t, \mathbf{X} = \mathbf{x}]$ (and thus $\mathbb{E}[Y|do(T = t)]$, the $ATE$) is to fit a model for $Y$ given $T$, and $\mathbf{X}$. These models are the basic building blocks of most methods for treatment effect estimation. We use the finite sample approximation of *ATE* by taking the average of *ITE*s. Following (Shalit et al., 2017; Lechner, 2001; Imbens, 2000; Schwab et al., 2020; Zhang et al., 2021), we make the following assumptions which are sufficient to guarantee the *identifiability* (Pearl, 2009) of treatment effects from observational data.

- **Ignorability:** This is also referred to as *no unmeasured confounding* assumption. For a given set of pre-treatment covariates, treatment is randomly assigned. Mathematically, in a binary treatment setting, conditioned a set of pre-treatment covariates $\mathbf{X}$, treatment $T$ is indepedent of the outcomes $Y^0, Y^1$ (i.e., $(Y^0, Y^1) \perp\!\!\!\perp T|\mathbf{X}$).

- **Positivity:** Treatment assignment for each individual is not deterministic, and it must be possible to assign all treatment values to each individual, i.e. $0 < p(t|\mathbf{x}) < 1 \ \forall t, \mathbf{x}$.
- **Stable Unit Treatment Value Assumption (SUTVA):** The observed outcome of any individual under treatment must be independent of the treatment assignment to other individuals.

**Neurosymbolic Program Synthesis:** Following (Shah et al., 2020), let $(\mathcal{P}, \theta)$ be a neurosymbolic program where $\mathcal{P}$ denotes the program *structure* and $\theta$ denotes the program *parameters*. $(\mathcal{P}, \theta)$ is differentiable in $\theta$ (see Appendix for an example neurosymbolic program). $\mathcal{P}$ is synthesized using a Context-Free Grammar (CFG). A CFG consists of a set of rules of the form $\alpha \rightarrow \sigma_1, \ldots, \sigma_n$ where $\alpha$ is a non-terminal and $\sigma_1, \ldots, \sigma_n$ are either non-terminals or terminals. Program synthesis starts with an initial non-terminal, then iteratively applies the CFG rules to produce a series of *partial structures*, viz. structures made from one or more non-terminals and zero or more terminals. These partial structures are considered as nodes in a program graph. The process continues until no non-terminals are left, i.e., we have synthesized a program. The leaf nodes of the resultant program graph contain structures that consist of only terminals. Let $\theta$ be the set of parameters of such a leaf node structure $\mathcal{P}$. Let $s(r)$ be the cost incurred in using the rule $r$ while generating a program structure. The structural cost of $\mathcal{P}$ is $s(\mathcal{P}) = \sum_{r \in R(\mathcal{P})} s(r)$, where $R(\mathcal{P})$ is the set of rules used to create the structure $\mathcal{P}$. In this paper, we set $s(r)$ to a constant real number for all production rules (e.g., $s(r) = 1 \ \forall r \in R(\mathcal{P})$). The program learning problem is thus usually formulated as a graph search problem, i.e., starting with an empty graph, the graph is expanded by creating new partial structures (internal nodes of the graph) and structures (leaf nodes of the graph). When searching for an optimal program, parameters of the program (and program structures) are updated simultaneously along with the synthesis of the programs (Shah et al., 2020).

For a synthesized program $(\mathcal{P}, \theta)$, we define $\zeta(\mathcal{P}, \theta) = \mathbb{E}_{(\mathbf{x}, t, y) \sim \mathcal{D}}[((\mathcal{P}, \theta)(\mathbf{x}, t) - y)^2]$ as the error/cost incurred by $(\mathcal{P}, \theta)$ in estimating treatment effects. The overall goal of neurosymbolic program synthesis is then to find a structurally simple program (that can also be human-interpretable) with low prediction error, i.e. to solve the optimization problem: $(\mathcal{P}^*, \theta^*) = \arg\min_{(\mathcal{P}, \theta)} (s(\mathcal{P}) + \zeta(\mathcal{P}, \theta))$. We now describe our methodology.

## 4 Neurosymbolic Treatment Effect Estimator: Methodology

The overall idea of our methodology is to design a Domain-Specific Language (DSL) for treatment effect estimation that is fairly general, followed by the use of the standard $A^*$ search algorithm to synthesize programs given observational data from a specific causal model. We begin by discussing the DSL we design, followed by the program synthesizer. Note that one could view each primitive of our DSL as modules of existing learning-based treatment effect estimators such as TARNet or CFRNet (Shalit et al., 2017). We also theoretically analyze the usefulness of the search-based neurosymbolic program synthesizer for the given task.

### 4.1 Domain Specific Languages for Treatment Effect Estimation

Since a program synthesizer requires as input a set of input-output examples, unsurprisingly, we can pose the problem of treatment effect estimation as the problem of mapping a set of inputs to corresponding outputs. Concretely, given observational data $\mathcal{D}$, the set of pairs $\{(t_i, \mathbf{x}_i)\}_{i=1}^n$ act as inputs and the set of outcomes $\{y_i\}_{i=1}^n$ act as outputs.

For simplicity, let $\mathbf{v}_i = [t_i; \mathbf{x}_i]$ (concatenation of treatment and covariates) denote the $i^{th}$ input. A synthesized program learns to estimate the potential outcomes for unseen inputs by learning a mapping between given input-output examples. To bring interpretability to synthesized programs and to leverage the inductive biases considered in treatment effect estimation literature, we develop a DSL (Table 2) based on well-known program primitives that have connections to ideas used in literature for treatment effect estimation (illustrated in Table 1) (We later state Propn 4.2 that guarantees the existence of a DSL for treatment effect estimation task). As discussed earlier, existing

| Regularizer/
Architectural Changes | Program Synthesis
Alternative |
|---|---|
| Two-head/Multi-head network
(Farajtabar et al., 2020) (Shi et al., 2019)
(Shalit et al., 2017) (Schwab et al., 2020) | `if − then − else`
`subset` |
| Pre-treatment selection,
Propensity Score Matching (Shi et al., 2019) | `subset` |
| IPM regularization
(Shalit et al., 2017) (Farajtabar et al., 2020) | `transform` |

Table 1: Connection between inductive biases in existing literature and the program primitives in the proposed DSL. This equivalence allows us to device a method that doesn't require additional regularizers or architectural changes.

treatment effect estimation methods introduce inductive biases into machine learning models either

A DSL for the Treatment Effect Estimation Task

$$\alpha := \texttt{if } \alpha \texttt{ then } \alpha \texttt{ else } \alpha \mid \texttt{transform}(\alpha, \mu, \sigma) \mid \texttt{subset}(\alpha, [\texttt{a..b}]) \mid \texttt{const(S)} \mid \odot (\alpha, \alpha) \mid \mathbf{v}$$

| Program Primitive | Description |
| --- | --- |
| 1. $\texttt{if } \alpha \texttt{ then } \alpha \texttt{ else } \alpha$ | Simple $\texttt{if} - \texttt{then} - \texttt{else}$ condition. To avoid evaluating conditions, and to enable back-propagation, we implement smooth approximation of $\texttt{if} - \texttt{then} - \texttt{else}$. |
| 2. $\texttt{transform}(\alpha, \mu, \sigma)$ | Transform the input vector $\alpha$ into $\phi(\alpha)$ as $\phi(\alpha) = \frac{\alpha - \mu}{\sigma}$ where $\mu$ and $\sigma$ are mean and standard deviation of observational data. Feed $\phi(\alpha)$ into a NN to get a real number as output. |
| 3. $\texttt{subset}(\alpha, [\texttt{a..b}])$ | Select a set of features from the start index $\texttt{a}$ (including) to end index $\texttt{b}$ (excluding) from the input $\alpha$. Other features are set to 0. Feed this vector into a NN to get a real number as output. |
| 4. $\texttt{const(S)}$ | Learn a set of constants of shape $\texttt{S}$. |
| 5. $\odot(\alpha, \alpha)$ | Parameterized algebraic functions (e.g., $\alpha_1 + \alpha_2, \alpha_1 * \alpha_2; \alpha_1, \alpha_2 \in \mathbb{R}$). |

Table 2: A DSL for the treatment effect estimation task in Backus-Naur form (Winskel, 1993) and its semantics. $\mathbf{v}$ represents input from $\mathcal{D}$. More details of each primitives is provided below.

through regularizers or through changes in NN architectures. On a similar note, one could view a DSL as a set of inductive biases based on learnable program primitives (Shah et al., 2020; Chaudhuri et al., 2021). The proposed DSL is based on inductive biases used in treatment effect estimation literature. We next describe the connections between program primitives in our DSL in Table 2 and inductive biases used in traditional treatment effect estimation methods.

**Connection between multi-head neural network architectures and** $\texttt{if} - \texttt{then} - \texttt{else}, \texttt{subset}$**:** Recall that, in treatment effect estimation, our goal is to estimate the quantity $\mathbb{E}[Y|T = t, \mathbf{X} = \mathbf{x}]$. If a single model is used to estimate both $\mathbb{E}[Y|T = 0, \mathbf{X} = \mathbf{x}]$ and $\mathbb{E}[Y|T = 1, \mathbf{X} = \mathbf{x}]$, it is often the case that $\mathbf{X}$ is very high dimensional and hence the treatment $T$, which is often one dimensional, may be discarded by the model when making predictions. This will result in the estimated treatment effect being biased towards 0 (Künzel et al., 2019). To account for this, two separate models can be used to estimate $\mathbb{E}[Y|T = 0, \mathbf{X} = \mathbf{x}], \mathbb{E}[Y|T = 1, \mathbf{X} = \mathbf{x}]$. However, this method suffers from high variance in estimated treatment effect due to limited data in treatment-specific sub-groups and selection bias (Shalit et al., 2017).

In order to mitigate this problem, (Shalit et al., 2017) and subsequent efforts by (Shi et al., 2019; Schwab et al., 2020; Farajtabar et al., 2020) use an NN architecture in which two separate heads are spanned from a latent representation layer to predict treatment specific outcomes and thus achieve better treatment effect estimate with lower variance. Such two-head NNs can be implemented using a combination of $\texttt{if} - \texttt{then} - \texttt{else}$ and $\texttt{subset}$ program primitives. For example, to implement a two-head NN architecture, a neurosymbolic program synthesizer can perform the following: $\texttt{if}$ $\alpha_1 = \texttt{subset}(\mathbf{v}, [0..1]) = 1$ ($\texttt{subset}(\mathbf{v}, [\texttt{a..b}])$ takes a vector $\mathbf{v}$ as input and returns a real number as output as explained later in this section), the program synthesizer executes $\alpha_2$ $\texttt{else}$ it executes $\alpha_3$ where $\alpha_2, \alpha_3$ are two different sub-structures that act as two heads of the overall architecture.

Note that each $\alpha$ in the primitive: "$\texttt{if } \alpha \texttt{ then } \alpha \texttt{ else } \alpha$" returns a real number and hence the output of "$\texttt{if } \alpha \texttt{ then } \alpha \texttt{ else } \alpha$" is also a real number. For e.g., as discussed above, $\alpha$ here can be either $\texttt{subset}(\mathbf{v}, [0..1])$ or $\texttt{transform}(\mathbf{v}, \mu, \sigma)$ too. Here both $\texttt{subset}(\mathbf{v}, [0..1])$ and $\texttt{transform}(\mathbf{v}, \mu, \sigma)$ take a vector $\mathbf{v}$ as input and return a real number as output.

To avoid discontinuities and to enable backpropagation, following (Shah et al., 2020), we implement a smooth approximation of $\texttt{if} - \texttt{then} - \texttt{else}$. For example, smooth approximation of $\texttt{if a} > 0 \texttt{ then b else c}$ can be written as $\sigma(\beta \cdot \texttt{a}) \cdot \texttt{b} + (1 - \sigma(\beta \cdot \texttt{a})) \cdot \texttt{c}$, where $\sigma$ is the sigmoid function and $\beta$ is a temperature parameter. As $\beta \to 0$, the approximation approaches the usual $\texttt{if} - \texttt{then} - \texttt{else}$. It is now easy to see that multi-head NN architectures can be implemented using multiple $\texttt{if} - \texttt{then} - \texttt{else}$ and $\texttt{subset}$ primitives. It is important to note that we do not hard-code/pre-define the network architecture. Instead, the program synthesizer learns to generate programs such that the primitives are composed in any order it deems to be effective in minimizing the loss value during training (see Sec. 5 and Appendix for examples).

**Connection between IPM regularization and** $\texttt{transform}$**:** To improve the results from two head NN architectures (e.g., TARNet), CFRNet (Shalit et al., 2017) proposes to use *IPM regularization* (e.g., Maximum Mean Discrepancy (Gretton et al., 2012), Wasserstein distance (Cuturi & Doucet, 2014)) on a latent layer representation. This enforces the encoded distribution of treatment ($p(\mathbf{x}|t = 1)$) and

control ($p(\mathbf{x}|t = 0)$) groups to be close to each other. Minimizing *IPM* between $p(\phi(\mathbf{x})|t = 1)$ and $p(\phi(\mathbf{x})|t = 0)$ (where $\phi$ is the learned representation) is then the same as ensuring that treatment and covariates are independent (i.e., $T \perp\!\!\!\perp \mathbf{X}$) thus mimicking RCTs (Shalit et al., 2017). To introduce this kind of inductive bias, we introduce a program primitive called $\texttt{transform}(\alpha, \mu, \sigma)$ that transforms a given input vector $\alpha$ into $\phi(\alpha)$ using two other vectors $\mu, \sigma$ as $\phi(\alpha) = \frac{\alpha - \mu}{\sigma}$, where $\mu, \sigma$ are mean and standard deviations of the observational data $\mathcal{D}$.

If $\texttt{transform}(\alpha, \mu, \sigma)$ is applied to the entire dataset $\mathcal{D}$, the transformed data now has mean $\mathbf{0}$ and standard deviation $\mathbf{1}$ (where $\mathbf{0}$ and $\mathbf{1}$ are vectors of 0s and 1s respectively). When the two subpopulations $p(\mathbf{x}|t = 0)$ and $p(\mathbf{x}|t = 1)$ are distributed similarly to $\mathcal{D}$ (which would also satisfy the *ignorability* assumption), the means and standard deviations of $p(\phi(\mathbf{x})|t = 0)$ and $p(\phi(\mathbf{x})|t = 1)$ will also be approximately equal to $\mathbf{0}$ and $\mathbf{1}$ respectively (note that $\phi(\mathbf{x}) = \frac{\mathbf{x} - \mu}{\sigma}$, and the input to $\texttt{transform}(\alpha, \mu, \sigma)$ is the vector $[\mathbf{t}, \mathbf{x}] \in \mathcal{D}$). Then, the Maximum Mean Discrepancy between $p(\phi(\mathbf{x})|t = 0)$ and $p(\phi(\mathbf{x})|t = 1)$ will go towards zero when matching the first two moments of $p(\phi(\mathbf{x})|t = 0)$ and $p(\phi(\mathbf{x})|t = 1)$ for treatment effect estimation. The transformed vector $\phi(\mathbf{x})$ is subsequently fed into a multi-layer perceptron to produce a real number as output, which is then fed into other program primitives.

$\texttt{transform}(\alpha, \mu, \sigma)$ appears to be similar to data standardization, a data pre-processing step. However, unlike the fixed architecture in traditional NNs, program synthesis has the flexibility to choose when to use $\texttt{transform}(\alpha, \mu, \sigma)$ (for e.g, $\mathbf{x}$ could be the output of the 3rd program primitive, not input). Besides, though the reason to introduce $\texttt{transform}(\alpha, \mu, \sigma)$ is to mimic IPM regularization (specifically Maximum Mean Discrepancy), it is evident from our DSL that the program synthesizer can use $\texttt{transform}(\alpha, \mu, \sigma)$ multiple times in a program (see Appendix for examples).

**Connection between pre-treatment covariate selection and $\texttt{subset}$:** Under the *ignorability* assumption, pre-treatment covariates are controlled to find estimates of treatment effects. For e.g., (Shi et al., 2019) controls pre-treatment covariates via controlling propensity score (Rosenbaum & Rubin, 1983). However, it is not required to control all the covariates in the input. In order to identify a minimal set of pre-treatment covariates to control, we use $\texttt{subset}(\alpha, [\texttt{a..b}])$ primitive. If we do not know which indices to select, multiple instances of $\texttt{subset}(\alpha, [\texttt{a..b}])$ can be used by assigning different values to $\texttt{a}, \texttt{b}$ in each instance. Program synthesizer then selects appropriate $\texttt{subset}(\alpha, [\texttt{a..b}])$ for some $a, b$. $\texttt{subset}(\alpha, [\texttt{a..b}])$ also helps to identify the most important dimensions from a given input/hidden representation as explained in the $\texttt{if} - \texttt{then} - \texttt{else}$ example earlier. Finally, the chosen vector is fed into a multi layer perceptron to produce a real number as output which will subsequently be used by other primitives.

The other two simple program primitives–$\texttt{const}(\texttt{S}), \odot(\alpha, \alpha)$–whose semantics are given in Table 2, are included for giving additional flexibility to the program synthesizer and combining various program primitives effectively to achieve better results. $\odot(\alpha, \alpha)$ takes two real numbers as inputs and returns a real number as output after performing the algebric operation $\odot$. Using the proposed DSL, we now present the algorithm to synthesize neurosymbolic programs that estimate treatment effects.

## 4.2 Neurosymbolic Program Synthesis for Treatment Effect Estimation

We use the $A^*$ informed search algorithm (Hart et al., 1968) to implement the proposed NESTER method. The heuristic function $h$ we use in our method is defined as follows. For any partial structure $\mathcal{P}(u)$ in a node $u$, NNs with adequate capacity (enough width and depth) are used to replace the non-terminals. The training loss of the resultant program $(\mathcal{P}(u), \theta(u))$ on $\mathcal{D}$ then acts as the heuristic value $h(u)$ at the node $u$ (Shah et al., 2020). Using this heuristic function, we run $A^*$ algorithm to find the programs that estimate treatment effects. We outline our approach in Algorithm 1. We now study the theoretical guarantees of neurosymbolic program synthesis in estimating treatment effects.

**Definition 4.1.** *(Admissible Heuristics (Harris, 1974; Pearl, 1984)) In an informed search algorithm, a heuristic function $h(u)$ that estimates the cost to reach goal node from a node $u$ is said to be admissible if $h(u) \leq h^*(u), \forall u$ where $h^*(u)$ is the actual/true cost to reach the goal node from $u$. $h(u)$ is said to be $\epsilon-$admissible if $h(u) \leq h^*(u) + \epsilon, \forall u$.*

**Proposition 4.1.** *In an informed search algorithm, let the cost of the leaf edge $(u_i, u_l)$ (edge connecting internal node $u_i$ to leaf node $u_l$) be $s(r) + \zeta(\mathcal{P}, \theta^*)$, where $\theta^* = \arg\min_\theta \zeta(\mathcal{P}, \theta)$ and $r$ is the rule used to create $u_l$ from $u_i$. If NNs $\mathcal{N}$ parameterized by their capacity (architecture width and height) are used to substitute the non-terminals in the partial structure of $u_i$, the resultant program's training loss is equal to the $\epsilon-$admissible heuristic value at the node $u_i$. Such an $\epsilon-$admissible*

*heuristic returns a solution whose path cost is at most an additive constant $\epsilon$ away from the path cost of the optimal solution (Shah et al., 2020).*

**Proposition 4.2.** *Given an $\epsilon$−admissible heuristic, for any trained 1-hidden layer NN $\mathcal{N}$ with $m$ inputs, $n$ hidden neurons, and one output, there exist a Domain Specific Language $\mathcal{L}$ such that the error/loss incurred by the synthesized program $(\mathcal{P}, \theta)$ is $\epsilon$−close to the error/loss incurred by $\mathcal{N}$ in approximating any continuous function.*

Proofs of the above propositions are in the Appendix. The universal approximation theorem (Hornik et al., 1989) states that we can increase the number of hidden layer neurons of a 1-hidden layer NN $\mathcal{N}$ to approximate any continuous function $f$ with a certain error, say $\hat{\epsilon}$. Proposition 4.2 states that there exists a neurosymbolic program $(\mathcal{P}, \theta)$ whose error in approximating $\mathcal{N}$ is $\epsilon$. Equivalently, there exists a neurosymbolic program $(\mathcal{P}, \theta)$ whose error in approximating $f$ is $(\epsilon + \hat{\epsilon})$. That is, if the relationship between treatment and effect is a continuous function, neurosymbolic programming is a viable candidate for estimating treatment effects.

---

**Algorithm 1:** NESTER using $A^*$

**Input:** Source node $u_0, Q \coloneqq \{u_0\}, f(u_0) \coloneqq \infty$

  **while** $Q \neq \emptyset$ **do**
    $v \leftarrow \arg\min_{u \in S} f(u)$
    $Q \leftarrow Q \setminus \{v\}$
    **if** *$v$ is goal node* **then**
      | return $v$
      **end**
    **else**
      create new partial architectures from $v$ (children of $v$) using DSL $\mathcal{L}$
      **foreach** *child $u$ of $v$* **do**
        | $h(u) \leftarrow \min_{\theta(u)} \zeta(\mathcal{P}(u), \theta(u))$
        | $f(u) \leftarrow s(\mathcal{P}(u)) + h(u)$
        | $Q \leftarrow Q \cup \{u\}$
        **end**
      **end**
  **end**

---

## 5 EXPERIMENTS AND RESULTS

We perform experiments to showcase the usefulness of NESTER in estimating treatment effects when coupled with our proposed DSL. Our code along with instructions for reproducibility of results is in the supplementary material. To permit interpretability, we limit the program depth to utmost 5 for the main experiments (see Appendix for experiments with other depths).

**Datasets and Baselines:** Evaluating treatment effect estimation methods requires all potential outcomes to be available, which is impossible due to the fundamental problem of causal inference. Thus, following (Shalit et al., 2017; Yoon et al., 2018; Shi et al., 2019; Farajtabar et al., 2020), we experiment on two semi-synthetic datasets–Twins (Almond et al., 2005), IHDP (Hill, 2011)–that are derived from real-world RCTs (see Appendix for details). For these two datasets, ground truth potential outcomes (a.k.a. counterfactual outcomes) are synthesized and available, and hence can be used to study the effectiveness of models in predicting potential outcomes. We also experiment on one real-world dataset–Jobs (LaLonde, 1986)–where we observe only one potential outcome. Each dataset is split 64/16/20% into train/validation/test sets, similar to earlier efforts.

We compare NESTER with Ordinary Least Squares with treatment as a feature (OLS-1), OLS with two regressors for two treatments (OLS-2), $k$-Nearest Neighbors ($k$-NN), balancing linear regression (BLR) (Johansson et al., 2016), Bayesian additive regression trees (BART) (Chipman et al., 2010), random forest (Breiman, 2001), causal forest (Wager & Athey, 2018), balancing neural network (BNN) (Johansson et al., 2016), treatment-agnostic representation network (TARNet) (Shalit et al., 2017), multi-head network (MHNET) (Farajtabar et al., 2020), Generative Adversarial Nets for inference of individualized treatment effects (GANITE) (Yoon et al., 2018), counterfactual regression with Wasserstein distance (CFR$_{WASS}$) (Shalit et al., 2017), Dragonnet (Shi et al., 2019) and multi-task Gaussian process (CMGP) (Alaa & van der Schaar, 2017).

**Evaluation Metrics:** For the experiments on IHDP and Twins datasets where we have access to both potential outcomes, following (Shalit et al., 2017; Yoon et al., 2018; Shi et al., 2019; Farajtabar et al., 2020), we use the evaluation metrics–*Error in estimation of Average Treatment Effect* ($\epsilon_{ATE}$) and *Precision in Estimation of Heterogeneous Effect* ($\epsilon_{PEHE}$). $\epsilon_{ATE}$ is a global measure in the sense that it measures the error in the estimation of average treatment effect in a population. $\epsilon_{PEHE}$ is a local measure as it operates on the error in the estimation of individual treatment effects. For the experiment on the Jobs dataset where we observe only one potential outcome per data point, following (Shalit et al., 2017; Yoon et al., 2018; Shi et al., 2019; Farajtabar et al., 2020), we use the metric *Error in estimation of Average Treatment Effect on the Treated* ($\epsilon_{ATT}$). Mathematical definitions and details of these metrics are provided in the Appendix. Following (Shalit et al., 2017; Shi et al., 2019; Yoon et al., 2018), we report both in-sample and out-of-sample performance w.r.t.

| Datasets → | IHDP | | Twins | | Jobs | |
|---|---|---|---|---|---|---|
| Metrics → | $\epsilon_{ATE}$ | | $\epsilon_{ATE}$ | | $\epsilon_{ATT}$ | |
| Methods ↓ | In-Sample | Out-of-Sample | In-Sample | Out-of-Sample | In-Sample | Out-of-Sample |
| OLS-1 | .73 ± .04 | .94 ± .05 | .0038 ± .0025 | .0069 ± .0056 | **.01 ± .00** | .08 ± .04 |
| OLS-2 | .14 ± .01 | .31 ± .02 | .0039 ± .0025 | .0070 ± .0059 | **.01 ± .01** | .08 ± .03 |
| BLR | .72 ± .04 | .93 ± .05 | .0057 ± .0036 | .0334 ± .0092 | **.01 ± .01** | .08 ± .03 |
| k-NN | .14 ± .01 | .90 ± .05 | .0028 ± .0021 | **.0051 ± .0039** | .21 ± .01 | .13 ± .05 |
| BART | .23 ± .01 | .34 ± .02 | .1206 ± .0236 | .1265 ± .0234 | .02 ± .00 | .08 ± .03 |
| R Forest | .73 ± .05 | .96 ± .06 | .0049 ± .0034 | .0080 ± .0051 | .03 ± .01 | .09 ± .04 |
| C Forest | .18 ± .01 | .40 ± .03 | .0286 ± .0035 | .0335 ± .0083 | .03 ± .01 | .07 ± .03 |
| BNN | .37 ± .03 | .42 ± .03 | .0056 ± .0032 | .0203 ± .0071 | .04 ± .01 | .09 ± .04 |
| TARNet | .26 ± .01 | .28 ± .01 | .0108 ± .0017 | .0151 ± .0018 | .05 ± .02 | .11 ± .04 |
| MHNET | .14 ± .13 | .37 ± .43 | .0108 ± .0008 | .0101 ± .0002 | .04 ± .01 | .06 ± .02 |
| GANITE | .43 ± .05 | .49 ± .05 | .0058 ± .0017 | .0089 ± .0075 | **.01 ± .01** | .06 ± .03 |
| CFR$_{WASS}$ | .25 ± .01 | .27 ± .01 | .0112 ± .0016 | .0284 ± .0032 | .04 ± .01 | .09 ± .03 |
| Dragonnet | .16 ± .16 | .29 ± .31 | .0057 ± .0003 | .0150 ± .0003 | .04 ± .00 | .04 ± .00 |
| CMGP | .11 ± .10 | .13 ± .12 | .0124 ± .0051 | .0143 ± .0116 | .06 ± .06 | .09 ± .07 |
| **NESTER** | **.06 ± .04** | **.09 ± .07** | **.0034 ± .0026** | .0063 ± .0033 | .06 ± .00 | **.02 ± .01** |

Table 3: Results on IHDP, Twins, and Jobs datasets. Lower is better. The best numbers are in bold. Second best numbers are underlined. Simple machine learning models, ensemble models, and neural network based models are separated using horizontal lines. See Appendix for further analysis on k-NN results.

$\sqrt{\epsilon_{PEHE}}, \epsilon_{ATE}, \epsilon_{ATT}$ in our results. The in-sample evaluation is non-trivial since we do not observe counterfactual outcomes (all potential outcomes) even during training.

From the results in Table 3, except w.r.t. in-sample $\epsilon_{ATT}$ score in Jobs dataset, NESTER either outperforms or is competitive with the best alternative methods. Our method has the flexibility to learn both complex models that are required for small and complex datasets such as IHDP (complex models such as CMGP outperforms simple models such as OLS on IHDP) and to learn simple models to solve large and simple datasets such as Twins and Jobs (OLS, $k-$NN often perform better on Twins, Jobs compared to complex models).

**Flexibility in Applying Inductive Bias and Program Primitives as Regularizers:** Inductive biases, a set of assumptions we make to solve a ML problem, have a significant impact on the ML model performance at test time (Mitchell, 1980). For a given task, inductive biases are chosen based on the intuition that a particular way of problem-solving is better than others.

These intuitions either come from domain knowledge or from data analysis. As discussed earlier, these inductive biases are implicit in program primitives in the proposed DSL. Using the proposed DSL, for each dataset, NESTER has the flexibility to: (i) Choose or not choose a specific program primitive; (ii) Decide order in which the program primitives are used; and (iii) Use a specific program primitive multiple times. This flexibility allows NESTER to use inductive biases differently for different datasets to perform better. Table 4 shows the best programs synthesized by NESTER for IHDP, Twins, and Jobs datasets. Unlike traditional fixed architectures (e.g., IPM regularization followed by two head network in CFRNet), NESTER synthesizes path flows (equivalent to different architectures) to solve each dataset. Additional experimental details including analysis on the depth of synthesized programs, impact of the choice of DSL are provided in the Appendix.

| IHDP |
|---|
| `if subset(v, [0..1])` |
| `then transform(v, μ, σ)` |
| `else transform(v, μ, σ) )` |

| Twins |
|---|
| `subset(v, [0..|v|]))` |

| Jobs |
|---|
| `if subset(v, [0..|v|])` |
| `then subset(v, [0..|v|])` |
| `else subset(v, [0..|v|])` |

Table 4: Sample programs learned by NESTER. $|\mathbf{v}|$ = size of vector $\mathbf{v}$.

## 6 CONCLUSIONS

This paper presents a new neurosymbolic programming approach for treatment effect estimation, and also studies why neurosymbolic programming is a good choice for solving such a problem. By making an analogy between parameterized program primitives and the basic building blocks of machine learning models in the literature on treatment effect estimation, we propose a Domain Specific Language on which program synthesis is rooted. Our results and analysis on benchmark datasets with several baselines show the usefulness of the proposed approach. Exploring new program primitives corresponding to unexplored heuristics for the treatment effect estimation task is an interesting future direction.

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
