# OpenReview forum: "Estimating Treatment Effects using Neurosymbolic Program Synthesis"
_ICLR.cc/2023/Conference — Submitted to ICLR 2023_

### Official Review · Reviewer_K4Ej · 2022-10-25

**Confidence:** 4
**Correctness:** 3
**Technical Novelty And Significance:** 3
**Empirical Novelty And Significance:** 2
**Recommendation:** 3

**Clarity, Quality, Novelty And Reproducibility:**

The submission is clearly written, is of high quality, and appears to be novel and reproducible.

**Strength And Weaknesses:**

Strengths:
Overall the paper is well written and addresses an interesting and important topic in causal inference, effect estimation with unknown structure. Despite my concerns listed below, I see many advantages of the program synthesis approach to effect estimation for reasons that authors discuss. For example, existing methods that combine structure learning with effect estimation lack the interpretability, extensibility, etc. of the synthesis approach presented here. In other words, I am strongly in support of viewing causal models as code, and viewing effect estimation and structure learning as program synthesis.

In addition to the framing and presentation, the majority of the arguments the authors present are well supported by clear and thorough evidence. The proofs in the appendix appear to be correct and well presented, and the ablation studies are much appreciated.

Opportunities for improvement:
My largest concern with this submission, which is a significant one, is that it conceptually and empirically ignores the large literature on causal graph based structure learning. This is a problem for three reasons.

(1) While the authors make standard assumptions in effect estimation explicit, they do not make similarly standard assumptions in causal graph structure learning explicit. For example, do the authors assume faithfulness? See Spirtes et al for more details on these assumptions.

(2) The authors assume no latent confounders, which is sufficient to identify causal effects in the setting where graph structure (or causal program structure) is known apriori. However, non-identifiability is a serious concern when structure is unknown. As a simple example, without strong parametric assumptions (e.g. additive noise) the graphs X -> Y and X <- Y are likelihood equivalent (where here likelihood is a conceptual proxy for the loss used in the paper), but yield dramatically different effect estimates. By choosing a single model, the neurosymbolic approach underestimates the uncertainty in the induced causal effects. It is possible that this might not be an issue with sufficient restrictions on programs (e.g. outcome is always a descendent of treatment), but that must be showed rigorously.

(3) The authors compare empirically only against methods that assume a fixed structure, and where all covariate are pre-treatment. It is important to add two additional baselines. First, it would be useful to show the performance of all of the baseline effect estimation methods when their assumption about covariates being pre-treatment is satisfied. In other words, remove all mediators from their input. Second, it is very important to compare against a handful of baselines that also learn causal structure. One example is to run PC, GES, and/or MMHC (See Spirtes, et al) first, and then to run one of the existing baseline effect estimation methods using the learned structure. Another strong baseline would be to apply the IDA algorithm from Maathuis et al.

Spirtes, Peter, et al. Causation, prediction, and search. MIT press, 2000.

Maathuis, Marloes H., Markus Kalisch, and Peter Bühlmann. "Estimating high-dimensional intervention effects from observational data." The Annals of Statistics 37.6A (2009): 3133-3164.



**Summary Of The Paper:**

This submission presents a methodology for treatment effect estimation by synthesizing causal programs from data. They go on to prove that synthesized programs in the DSL are expressive enough to approximate any continuous function with a small epsilon bound. Finally, they show that the neurosymbolic program synthesis approach outperforms standard effect estimation that assume that the causal graph is fixed.

**Summary Of The Review:**

Overall this submission is a promising contribution to the literature. However, it is very important to address comparisons to structure learning based approaches that are prevalent throughout the causal inference literature.

---

### Official Review · Reviewer_Ykau · 2022-10-25

**Confidence:** 4
**Correctness:** 3
**Technical Novelty And Significance:** 2
**Empirical Novelty And Significance:** Not applicable
**Recommendation:** 3

**Clarity, Quality, Novelty And Reproducibility:**

The paper has clarity and reproducibility, however as I have explained above there needs to be additional clarity on the benefits of using this estimator in comparison to existing estimators.

**Strength And Weaknesses:**

Strengths: Estimation of Treatment Effects identified via backdoor-adjustment hasn't been explored using neurosymbolic programs before, so there is novelty here. However, please see the discussion of weakness below.

Weakness: When it comes to treatment effect estimation, since we are reasoning with interventions that are not observed and we don't have a ground truth for - it is highly important to be able to provide statistical inference guarantees. Specifically, consistency (I do see a proposition that talks about approximating continuous functions), a favourable rate such as sqrt-n, and valid uncertainty quantification are all desired from a statistical estimator. The theory of influence functions provides us with an Augmented Inverse Probability Weighted (AIPW)estimator which provides sqrt-n rates while allowing for flexible estimation of nuisance parameters using non-parametric estimators. So I don't understand how this proposed neurosymbolic approach fits in with existing approaches for estimating treatment effects. Either this estimator offers lower variance than AIPW, in which case I would like to see a proof. Or if the results of the neurosymbolic estimator are non-asymptotic, I would like to see proofs there as well.

All in all this paper needs to engage more with the existing literature on treatment effect estimation in order to explain where it improves existing well-established estimators.

**Summary Of The Paper:**

This paper provides a new neurosymbolic treatment effect estimator for estimating treatment effects when the backdoor adjustment formula can be used for identification. Casting the treatment effect estimation problem as a neurosymbolic program is novel, and resulting theory around the ability to approximate continuous functions is interesting. Finally, performance is demonstrated on experimental datasets.

**Summary Of The Review:**

Based on the details explained in the weaknesses section, I do not recommend this paper be accepted. However, I'd be happy to change my mind if the authors can satisfactorily address my questions.

---

### Official Review · Reviewer_iRvp · 2022-10-28

**Confidence:** 3
**Correctness:** 3
**Technical Novelty And Significance:** 3
**Empirical Novelty And Significance:** 3
**Recommendation:** 8

**Clarity, Quality, Novelty And Reproducibility:**

The paper is mostly clear and a pleasure to read. The one issue is related to the Propositions, where the cost of a rule wasn't defined (apologies if I missed it in the paper, but I couldn't find it the main text nor in the appendix).

The evaluation was well done as far as I can tell (I am not a specialist in causal inference) with many baselines from the literature.

The work also seems to be novel.

The code was provided with the submission -- I am assuming then that it should be easy to reproduce the results.

**Strength And Weaknesses:**

Strength

The use of neurosymbolic programming for solving the treatment effect estimation problem is novel. The way the ideas are presented in the paper allows someone who isn't familiar with the topic to understand enough of the problem to appreciate the results. Overall the paper is very pleasing to read. I thank the authors for it!

I also appreciated the connection between each functionally of the DSL with existing inductive biases from the literature.

Finally, the results are good, perhaps not very strong, but good. The standard deviation in some of the columns are somewhat large and makes one wonder how significant some of the results are, but the results for "Jobs" (out-of-sample) are strong with a clear difference between the proposed method and baselines.

Weaknesses

Although the paper is very pleasing to read, it is sloppy in the definitions around the propositions of the paper. While I am not absolutely certain (I hope to better understand this issue with the authors' response), I suspect that the propositions do not hold. Here is why.

Both propositions build on the fact that the heuristic is epsilon-admissible, but the paper never defined (I apologize if I missed this definition) what is the cost of a rule, $s(r)$. This is important because the A* search uses two components: g(n) and h(n) for a given node in the A*'s search tree. The first is the cost traversed from the root of the tree to node n, while the second is the estimated cost-to-go from n to a goal state. The paper defines $\zeta(P, \theta)$ for the program $(P, \theta)$ as the squared loss of what the program produces and the true outcome $y$.

While $\zeta(P, \theta)$ makes sense in the context of guiding the synthesis (i.e., partial programs with $\zeta(P, \theta)$-values smaller will be explored first in search), I don't see how it fits the A* framework because $s(r)$ would have to be defined as something like "the squared error of each rule used to build the program". However, I don't see how this is possible as one can't really assign a squared error to a rule; it would make sense to assign an error to a program, but not a rule used to build a program.

There is a chance that $s(r)$ and $\zeta(P, \theta)$ represent two quantities that can't be compared and thus $\zeta(P, \theta)$ cannot be an $\epsilon$-admissible heuristic function for the problem. This would invalidate both propositions of the paper.

My lack of understanding $s(r)$ is what is holding me from recommending acceptance of the paper. Authors, please help me understand what $s(r)$ is and I will probably be able to connect the points and understand why $\zeta(P, \theta)$ is $\epsilon$-admissible.

**Summary Of The Paper:**

This paper introduces a neurosymbolic method for dealing the "treatment effect estimation" problem. The method is able to synthesize programs in a DSL that was carefully constructed to carry the inductive biases of previous neural approaches for solving the same problem. The programs combine the functionalities of the DSL with neural networks.

The process in which the programs are synthesized is to use approximations of partial programs (programs with holes) where the holes are replaced by neural networks.

Empirical results on three different data sets show that the neurosymbolic method is either superior or competitive with other methods in the literature.

**Summary Of The Review:**

Interesting paper, easy to read, and with good results. The method seems to be novel and some of the ideas presented in this paper can possibly be used to solve other problems as neurosymbolic programs are very general.

The propositions could be false and I am hoping the authors will be able to clarify this issue.

---

### Decision · Program_Chairs · 2023-01-20

**Decision:**

Reject

**Justification For Why Not Higher Score:**

I have a hard time appreciating what is gained by this extra combinatorial search. So much is well-understood in causal effect estimation that by now the bar should be high. For instance, many methods are modular and can easily build on stacking to just take a kitchen sink approach to combine all sorts of machine learning methods as building blocks - for instance, the SuperLearner of https://ctml.berkeley.edu/resources/software is very popular and simple to use. It seems unlikely how much overfitting is happening in the experiments. Also, some of the metrics are not correct, e.g. in Section 8.1, eq. 10 epsilon_EPHE is incorrectly defined (it is should not make use of (Y_i^1 - Y_i^0)^2 which among other things depend on the covariance of potential outcomes, a quantity which is not only unidentifiable even with experimental data, but it is not taken into consideration when defining the causal estimates). It is most definitely not the metric that the other competitors in the benchmark claim they can minimise.

**Justification For Why Not Lower Score:**

N/A

**Metareview: Summary, Strengths And Weaknesses:**

The paper makes use of neurosymbolic program synthesis (based on combinatorial search) to produce models for causal effect estimation.

Strengths: provides an unification of some ideas for developing deep learning architectures in the context of causal effect estimation.

Weaknesses: I have a hard time appreciating what is gained by this extra combinatorial search. So much is well-understood in causal effect estimation that by now the bar should be high. For instance, many methods are modular and can easily build on stacking to just take a kitchen sink approach to combine all sorts of machine learning methods as building blocks in a single shot - for instance, the SuperLearner of https://ctml.berkeley.edu/resources/software is very popular and simple to use. It seems unlikely how much overfitting is happening in the experiments. Also, some of the metrics are not correct, e.g. in Section 8.1, eq. 10 epsilon_EPHE is incorrectly defined: it is should not make use of (Y_i^1 - Y_i^0)^2 which among other things depend on the covariance of potential outcomes, a quantity which is not only unidentifiable even with experimental data, but it is not taken into consideration when defining the causal estimand of interest. It is most definitely not the metric that the other competitors in the benchmark claim they can minimise.

I acknowledge I read all reviews and rebuttals in detail, and I caught some misunderstandings in a few passages in the reviews. I appreciate the answers from the authors correctly point out a few of these misunderstandings. But the main point remains.